# Investigating the ME/CFS experience through qualitative analysis of memorial entries

Zoe Sirotiak[1,2]*, Hailey J. Amro[2]

1 Department of Kinesiology, Iowa State University, Ames, Iowa, United States of America, 2 Department of Psychology, Iowa State University, Ames, Iowa, United States of America

* zmsiro@iastate.edu

## Abstract

Myalgic encephalomyelitis/chronic fatigue syndrome (ME/CFS) is an impairing chronic condition characterized by exhaustion and worsening symptoms following exertion, often accompanied by pain, sleep issues, and cognitive issues. Historically, ME/CFS was not considered to be linked to mortality, however, more recent studies have questioned this assumption. The National Chronic Fatigue and Immune Dysfunction Syndrome (CFIDS) Foundation maintains a memorial list consisting of deceased individuals who had ME/CFS. This secondary qualitative thematic analysis analyzed 505 entries on the National CFIDS Foundation memorial list, inductively developing a codebook from the publicly available memorial records. Two coders independently coded each entry before meeting to develop themes that incorporated the understanding of each coder. Themes emerged within four societal levels: systemic neglect and institutional failure; clinical neglect and failures; social disconnection and advocacy; and personal burden and quality of life. Describing systemic neglect and institutional failure, entries recounted a lack of acknowledgement by health, insurance, and disability authorities, as well as a lack of investment in research and treatment of ME/CFS at the federal level. Negative healthcare experiences included misdiagnosis and misattribution of symptoms, dismissal, inadequate knowledge and experience with treating ME/CFS, and the recommendation of unhelpful treatments. The disbelief and misattribution by acquaintances described in the entries contributed to feelings of social isolation, leading some to turn to advocacy work and support groups. Entries also described the individual impact of the condition, including functional impairments, the impact of symptoms, management strategies, financial stress, and mental health symptoms. Some deaths were directly and indirectly attributed to ME/CFS by individuals with ME/CFS and their acquaintances. This analysis provides a glimpse of the lived experience as well as death of individuals with ME/CFS through the lens of acquaintances of the deceased, emphasizing the substantial impact of the condition.

**Data availability statement:** Data supporting this paper is available at https://www.ncf-net.org/memorial and in this paper and its Supporting Information files.

**Funding:** The author(s) received no specific funding for this work.

**Competing interests:** The authors have declared that no competing interests exist.

## Introduction

Myalgic encephalomyelitis/chronic fatigue syndrome (ME/CFS) is a biological illness, that impacts several different body systems, resulting in chronic disruptions to daily activities [1]. The condition can present with varying degrees of severity and heterogeneity in symptom profiles [2]. Among the most common symptoms experienced include fatigue, post-exertional malaise, cognitive impairments, chronic pain, sleep problems, and autonomic symptoms [3]. Previous research has divided symptoms of ME/CFS into clinically relevant subgroups, including inflammatory, pain, neurocognitive, and autonomic symptom subgroups, highlighting the heterogeneous impact on affected individuals [4]. ME/CFS has historically been underfunded and under-researched [5,6], contributing to the lack of objective measures for diagnosis, effective treatment protocols, and understanding of underlying mechanisms [7].

Recent literature has highlighted the potential utility of an ecological systems approach to understanding the multidimensional impacts of ME/CFS [8]. The ecological systems model asserts that an individual's experience is shaped by various levels of impact with the individual at the core. Surrounding the individual is the microsystem: the individual's immediate environment, the mesosystem: interactions between two or more microsystems, the exosystem: larger social structures, the macrosystem: cultural norms, social policies, and other systemic factors, and the chronosystem: the influence of time on an individual's experience [9,10] Some have asserted that a systems approach could allow for increased accuracy in diagnosing and understanding of the patient experience [11]. Research has highlighted several relevant systemic factors that contribute to understanding the experience of ME/CFS, including consistent negative experiences with the healthcare system [12], significant underdiagnosis of the condition [13], stigmatization [14], and lack of specialized care [15]. ME/CFS has been associated with significant limitations in health-related quality of life [16], indicating the importance of considering the lived experience of individuals with the condition, including the barriers and challenges faced. Similarly, this ecological model has been utilized to enhance health literacy and build understanding of other illnesses such as acute SARS-CoV-2 infection, indicating the importance of understanding the impact of the micro-, meso-, exo-, and macro- systems of the ecological model on an individual's illness experience [17].

Despite the substantial impacts of ME/CFS on quality of life and functional status [18,19], ME/CFS has historically not been considered to significantly alter an individual's lifespan [20]. However, more recent studies have questioned this assertion, suggesting that some with ME/CFS experience earlier mortality [21–23]. Jason et al. (2006) conducted an analysis of 166 memorial entries on the National CFIDS Foundation website through the summer of 2003 [22]. The authors conducted a quantitative analysis of the memorial entries and reported on causes of death and demographic considerations in deaths of those with ME/CFS. This study suggested that ME/CFS may be associated with earlier mortality for some individuals, compared to the general population [22]. A recent updated analysis reinforced these findings in a larger sample of 505 deceased individuals with ME/CFS, finding cancer, cardiovascular problems, and suicide as among the frequently reported causes of death

among individuals with ME/CFS [23]. This updated analysis also found a substantial proportion (28.3%) of deaths among individuals with ME/CFS attributed to ME/CFS or complications of the condition through narrative report [23], reinforcing the idea that individuals with ME/CFS and those around them may attribute death directly or indirectly to ME/CFS itself.

In a separate study, family, friends, and caregivers of deceased individuals with a ME/CFS diagnosis were surveyed regarding the limitations, symptoms, functional abilities, and circumstances of death [21]. This study indicated a significantly increased risk of earlier all-cause mortality compared to the overall United States population [21]. Despite these investigations indicating the importance of considering mortality in the context of ME/CFS, qualitative analyses of mortality data among individuals with ME/CFS are rare. Therefore, this study aims to conduct a qualitative analysis of the data presented within the National CFIDS Foundation memorial list. This analysis takes an inductive approach to uncover nuances within the memorial data and seek a deeper understanding of the lived experiences and deaths of those with ME/CFS. The authors hope to incorporate an ecological systems perspective to acknowledge the multilevel effect of ME/CFS on the functioning of patients [24].

## Materials and methods

The National CFIDS Foundation memorial page [25] is a list of deceased individuals who lived with ME/CFS or chronic fatigue and immune dysfunction syndrome (CFIDS). As ME/CFS, CFIDS, chronic fatigue syndrome (CFS), and myalgic encephalomyelitis (ME) were all treated interchangeably in the memorial list, the condition will be referred to as ME/CFS, aligning with Centers for Disease Control and Prevention convention [1]. Family members, friends, or acquaintances can submit information about deceased individuals who had ME/CFS to the memorial list. Information provided varies by entry but frequently provided information includes characteristics of the deceased individual (e.g., age at death, sex, year of death, geographical location), the individual's experience with ME/CFS, as well as the cause and circumstances of death. The memorial list was downloaded in October 2024, with a total of 505 entries. Individuals included on the memorial list had an average age of 52.5 years (SD = 16.7) with an age range of 14–96 years. Most participants were female (73.0%), lived in North America (68.9%), and had a year of death in the 2000s or 2010s (73.3%). Detailed demographics of the sample have previously been reported [23]. The National CFIDS Foundation memorial page is publicly accessible [25], and therefore Institutional Review Board approval was not required for this analysis.

A thematic analysis using inductive coding strategies [26,27] was performed to analyze entries on the memorial list. Two researchers, each with experience in coding and interpreting qualitative data, were involved in the coding process. First, the coders familiarized themselves with the data, reading through each of the entries and taking notes regarding potential codes as well as patterns demonstrated in the entries. Coders also identified areas of confusion or lack of clarity in their own coding framework to prepare for codebook finalization in discussion. The coders then met to establish a codebook, discussing the meanings and distinctions of the codes developed. Coders shared their code list and framework, exploring similarities and differences between their coding frameworks. Discussion between coders included identifying foundational codes based on concepts that were expressed repeatedly, as well as subcodes. The coders also explored how and why interpretations of some codes may differ between coders or be unclear, resulting in clarifying and defining these codes. The coders engaged in reflexivity throughout the coding and thematic analysis process, examining their assumptions and potential biases. The coders considered how their educational background, professional training, personal experiences, and underlying assumptions may have influenced their interpretation of the memorial narratives and resulting code development. Discussion of themes between the coders was designed to allow consideration of diverging perspectives, resulting in a more comprehensive understanding of the narratives. Saturation was not assessed as this was a secondary data analysis. Saturation has been criticized as a heterogenous and inconsistently applied concept [28], further supporting its omission from this analysis.

Each of the coders independently coded each of the entries before meeting again to discuss emerging initial themes. Each code was dichotomously coded as either "present" or "absent" within each memorial narrative utilizing an Excel

document. Each of these coded memorial entries are shown in S1 Table, with specific participant names removed for privacy purposes. The coders identified associations between codes and resulting themes, as well as associations between themes. Interrater reliability was not assessed, aligning with recommendations for thematic analyses [29]. Instead, the coders considered individual interpretations and worked together to develop themes that considered the understanding of both coders. Analytic trustworthiness was promoted through coder discussions in which coders compared their interpretations and reflected on how their background and perspective may influence coding decisions and conclusions. After these themes were identified and named, the researchers read through selections of the entries to ensure that these themes accurately and adequately described the ideas and experiences communicated in the entries. The themes were then finalized with descriptive quotes identified to illustrate the emerging themes. Emerging themes and subthemes are displayed in Table 1, and codes and subcodes are shown in S1 Table.

While all entries analyzed are publicly available [25], full names have been omitted to protect privacy. All quotes are provided verbatim, including original spelling, punctuation, and grammatical errors, to allow for the voice of the individual or their acquaintance to be accurately shared. Although these data were publicly available, memorial narratives are inherently sensitive and personal, and we approached our analyses with care. We chose to remove names and other specific information (i.e., city, insurance company name) to reduce direct identifiability while retaining gendered pronouns (he/she) to allow narrative coherence and respecting the expressive purpose of the original text.

## Positionality

The authors acknowledge the impact of their identities and experiences in shaping their interpretations and conclusions. The first author is a white, cisgender American woman. She holds a bachelor's degree in kinesiology and psychology, a Doctor of Physical Therapy degree, a master's degree in kinesiology, and is pursuing a PhD degree with majors in immunobiology, kinesiology, and psychology. Her academic background informs her focus on the interconnections between physical and mental health in the context of chronic health conditions, seeing all health conditions as not dichotomously physical or mental health issues. She acknowledges her social privilege due to her education and academic training, and she appreciates her lived experience limitations regarding personal experience of underprivileged social identities.

**Table 1. Description of emerging themes and subthemes.**

| Theme | Subthemes | Description |
|---|---|---|
| Systemic neglect and institutional failure | Lack of acknowledgement | Insurance, disability, larger healthcare, and other systems fail to validate ME/CFS as a legitimate condition |
| | No investment | Minimal funding or institutional support reinforces systemic neglect |
| Clinical neglect and failures | Negative healthcare experiences | Misattribution, dismissal, and inappropriate treatment recommendations |
| | Lack of knowledge | Inaccurate death coding, ineffective treatments, and general medical ignorance about ME/CFS |
| Social disconnection and advocacy | Impact on relationships | Social isolation and loss of connection with peers and family |
| | Disbelief and advocacy | Struggles for validation leading to involvement in support groups and awareness efforts |
| Personal burden and quality of life | Functional impairments | Challenges with activities of daily living, employment, housing, and caregiving |
| | Illness experiences | Chronic pain, comorbidities, and symptom management |
| | Life satisfaction | Emotional distress, financial hardship, and in some cases, suicidal ideation or death |

She has lived with ME/CFS for approximately six years, contributing lived experience and a patient perspective that may influence how she interprets memorial narratives. The second author is a cisgender American woman with mixed Arab heritage. She holds a bachelor's degree in psychology, a Master of Social Work degree, and is pursuing a PhD degree in Counseling Psychology. Her academic background and clinical social work experience inform her focus on an ecological systems framework, which emphasizes the impact of external factors on an individual's lived experiences. Her background also informs her focus on highlighting the lived experiences of and addressing the barriers faced by marginalized communities such as those with chronic illness. She acknowledges her social privilege stemming from her academic and social backgrounds, as well as her lack of personal experience in living with chronic illness. She made conscious efforts to check existing biases while interpreting the memorial narratives; however, her held identities may ultimately influence her interpretation of the data.

## Results

All 505 entries on the National CFIDS Foundation memorial page were coded, with these codes informing the theme-generating process. The authors identified themes emerging at varying levels of an individual's experience: the individual level, the interpersonal level, the community level, and the systems level. Four themes emerged within each of these areas. These primary themes at each level of experience include personal burdens and quality of life, social disconnection and advocacy, clinical neglect and failures, and systemic neglect and institutional failures, with each theme having additional subthemes. Personal burdens and quality of life were further characterized by subthemes of functional impairments, individual illness experiences, and varying degrees of life satisfaction. Social disconnection and advocacy were further clarified by illness impacts on interpersonal relationships, disbelief from others, and illness advocacy. Lack of knowledge and negative health experiences with healthcare providers are subthemes of the larger clinical neglect and failure theme. Lastly, the lack of acknowledgment and lack of investment from larger systems subthemes were characteristic of the systemic neglect and institutional failure's primary theme. Each of these themes also correspond to one or more intersecting levels of the ecological model. The identified levels, themes, and subthemes emerging are described below.

### Systemic neglect and institutional failure

Many entries expressed substantial frustration with the systems and institutions perceived as obstructing progress in understanding, diagnosing, and treating ME/CFS. National and local health organizations were criticized for stalling advances in research and treatment. Insurance companies were another source of exasperation, viewed as negligent and lacking empathy for those living with ME/CFS. The challenges of the legal disability process were another common theme, with entries describing the disability system as overly complex and emotionally taxing, further contributing to the burden of living with ME/CFS.

   3.1.1.  **Lack of acknowledgement.**  A recurring theme was the consistent lack of recognition and respect from health institutions and authorities, as they failed to validate ME/CFS as the serious and debilitating medical condition experienced by those described in the entries:

> "(She) *had found some help at the only hospital in* (county) *that treats CFIDS/ME but was denied funding by her local health authority for any further treatment and her health deteriorated dramatically*" [25].

Misunderstanding by healthcare systems was common, complicating management of an already poorly understood and stigmatized condition:

> "*The benefits agency made life hard, and the hurdles of having a disease so misunderstood left her depressed in addition to her devastating symptoms*" [25].

Others argued that the healthcare system held other priorities, often to the detriment of individuals with ME/CFS:

*"I can't go anywhere and don't have a moment free of pain. I'm not so much depressed as I am angry because the medical profession is too wrapped up in saving money (rather than) people"* [25].

The disability system was yet another challenge reported. Several individuals had negative experiences during their attempts to obtain disability benefits, often involving extensive and costly legal battles:

*"(She) spent years ferociously fighting her long-term disability company. When (company) sought to dismiss her charges when she claimed they treated "CFS" differently, a judge ruled that her ADA (disability claim) was valid. She publicly accused (company) of racketeering, extortion, wire fraud, collusion and attempted murder. She eventually settled for a large lump sum"* [25].

### 3.1.2. Lack of investment.
Entries also voiced frustration over the lack of institutional support and funding for ME/CFS treatment and research, leading to a sense of hopelessness regarding future improvements. Some individuals attempted to engage in advocacy efforts, often without notable success:

*"(He), meanwhile, began a petition online, along with the (foundation), imploring the government to recognize ME. It was still online with thousands and thousands of signatures when our (governmental organization) responded by eliminating the term of "myalgic encephalomyelitis" from their diagnostic code"* [25].

Others shared a similar experience, reporting a pattern of systemic dismissal that reinforced feelings of neglect and hopelessness:

*"She went to the (governmental organization) in 1982 for a diagnosis but was given none. In 1986, she got a copy of her records that indicated she had "Chronic Epstein-Barr Virus," now known as CFS. A few days before her death, she told a friend that this was being covered up from "very high up""* [25].

Although some individuals participated in advocacy at the federal level such as hearings and advisory committees, most reported little to no tangible outcomes, furthering frustration:

*"She gave testimony to the federal (committee) just months before her death that mentioned this Memorial List she has now joined, "They died because the (government) does not care to help...""* [25].

Several argued for more research funding dedicated to the understanding and treatment of ME/CFS, noting the importance of scientific advancement in promoting condition management:

*"It bears out the serousness of M.E. and it does emphasise how important it is that more research is carried out. It can strike the fittest of people. It just added to (her) frustration that she couldn't do what she used to do"* [25].

### 3.1.3. Clinical neglect and failures.
Despite the need for healthcare support for individuals with poorly understood conditions such as ME/CFS, many entries described struggles to access appropriate care and treatment. Many reported limited recognition of ME/CFS as a legitimate and serious illness with few effective management strategies. A lack of research resources devoted to investigating the mechanisms and treatments of ME/CFS contributed to clinical barriers to receiving effective care.

**3.1.4. Negative healthcare experiences.** Many entries highlighted negative interactions with the healthcare system, contributing to feelings of frustration and hopelessness about their health and the medical system's ability to address their symptoms. Dismissal by healthcare providers was another common theme, with some individuals avoiding medical care entirely due to poor past experiences:

*"Her experiences with physicians who dismissed her complaints made her avoid them as much as possible and she was taken to a hospital by ambulance only when her pain became unbearable"* [25].

Others described outright refusal of care, indicating a broader discomfort among healthcare professionals in working with and treating individuals with ME/CFS:

*"A severely progressive patient, she was told to find another physician by her* (state) *specialist the last year of her life when he no longer wanted to treat CFIDS/ME patients"* [25].

Misattribution of symptoms was also noted, with healthcare providers attributing ME/CFS symptoms to unrelated conditions or psychological causes:

*"She was told, repeatedly, that her illness was "all in her head" and was sick for 1,115 working days"* [25].

*"As she worsened with agonizing symptoms, several physicians believed she had something psychological wrong with her"* [25].

These misattributions sometimes led to inadequate or even harmful treatments and recommendations:

*"(He)* died… after being told, for 20 years by his GP, that his illness was 'all in his head'. The GP refused to sanction any care for him when his wife was admitted for a long hospital stay. The doctor stated that he could look after himself if he chose to as, according to him, ME did not exist. (He) *was too sick to feed himself nor could he even get himself a drink. He died with his bones sticking out, looking like someone from a concentration camp. His death certificate lists ME as the cause of death"* [25].

*"A talented writer,* (she) *was reported to have died "from chronic fatigue syndrome and anorexia". The latter condition was given by a consulting pathologist since* (she) *was bedbound, too weak to speak and had to have her food liquefied in order to eat. He was, of course, wrong just as others were wrong when they had, earlier, put her in a mental ward…"* [25].

**3.1.5. Lack of knowledge.** The widespread lack of experience and understanding of ME/CFS among healthcare providers was frequently noted. Several entries reported an unclear cause of death, complicated by the complexity of ME/CFS symptoms and outcomes. In some cases, an overdose was listed as the cause of death, though it was unclear whether the overdose was accidental or intentional. Other entries directly questioned the official cause of death:

*"Cause of death: gunshot wound. Suicide has been questioned by friends"* [25].

Some individuals encountered providers who directly denied the legitimacy of ME/CFS:

*"Her doctor was one who didn't believe in CFIDS/ME and would not prescribe the necessary pain medication"* [25].

*"When she went to (city) to see physicians who did not believe her illness existed, she took her own life in their building"* [25].

Other providers were not experienced working with individuals with ME/CFS, leading to poor treatment:

*"She had thought of suicide as her suffering became much worse but didn't act fast enough and she spent her last years in a nursing home where, again, she faced staff that had no knowledge what she was suffering from"* [25].

In other instances, treatment itself was actively harmful:

*"After a neurologist finally correctly diagnosed her, she still received both CBT (cognitive behavioral therapy) and GET (graded exercise therapy) that worsened her condition until she died"* [25].

*"… he collapsed and died as he was leaving the… gym… after following an exercise regime he was advised would help him with ME."* [25].

Delays in diagnosis and incorrect diagnoses were also common, leading to inappropriate treatments and chronic diagnostic uncertainty:

*"Years ago, (he) was misdiagnosed with multiple sclerosis and with an Alzheimer's-like condition until he was finally diagnosed with CFIDS/ME"* [25].

### Social disconnection and advocacy

Entries detailed the substantial impact of ME/CFS on social relationships, affecting the individual's daily experience and sense of belonging in their social groups and environment. Some social connections dismissed the existence and impact of ME/CFS, impairing relationships. Individuals responded to these struggles in varying ways, with some entries noting social disconnection as a precursor to poor mental health and sometimes suicide. Other individuals turned to advocacy efforts, aimed at improving acknowledgement of and treatment for ME/CFS.

**3.2.1. Impact on relationships.** Entries described the substantial impact that ME/CFS symptoms had on relationships:

*"She was confined to one room and could not tolerate noise or light. Her window blind remained closed. She tried to stay in touch with friends until she could no longer as her pain and other symptoms worsened a great deal"* [25].

Some individuals experienced substantial physical limitations due to ME/CFS symptoms, further contributing to social isolation:

*"(He) was a quiet man with dignity who often felt the profound isolation of myalgic encephalomyelitis while living in his remote country town"* [25].

*"(He) suffered from extreme pain in addition to numerous other symptoms and social isolation"* [25].

Several individuals noted the breakdown of relationships due to the impact of their symptoms:

*"Her husband divorced her… when doctors treated her for a mental disease and the medications they prescribed worsened her"* [25].

*"He also was divorced after the illness ruined his life"* [25].

However, other entries noted that acquaintances had also been diagnosed with ME/CFS, indicating connections within the ME/CFS community:

*"He had been employed at a workplace where seven other people now suffer from CFIDS/ME"* [25].

*"Her son and two grandchildren also developed the disease"* [25].

**3.2.2. Disbelief and advocacy.** Many entries described disbelief in ME/CFS by social acquaintances, limiting social support:

*"When she became sick, her own family was unsupportive and dismissed her as a mental patient"* [25].

*"She had suffered with CFIDS/ME for years in* (state) *with little or no support from her family who did not "believe" she was ill"* [25].

With limited support from existing social connections, some entries described looking elsewhere for validation and social support. Several entries noted joining or founding support groups for individuals living with ME/CFS:

*"She created her own support group via Instagram and had more than 2,000 followers"* [25].

Some engaged in advocacy, volunteering their time to educate the public and healthcare professionals about ME/CFS, advocating for increased recognition and support for individuals with ME/CFS at the systems level:

*"(She) was a staunch advocate for CFIDS/ME, was interviewed by* (media outlet) *and spoke at many conferences to educate physicians"* [25].

*"(Her) symptoms became so severe after a few years that she became bedridden yet most physicians felt she had a mental disease. She joined her mother and boyfriend advocating for funding as she believed there was an effective treatment and or cure somewhere but her hopes were dashed each time"* [25].

*"At a conference in* (city)*, he gave speech that, as was* (his) *style, didn't mince words, as he said, "We are sick, often deathly ill and we are NOT fatigued... change the God Damn Name!""* [25].

**3.2.3. Personal burden and quality of life.** Entries described the lived experience of ME/CFS, noting limitations across many areas of life, including activities of daily living, employment, housing, caregiving, and hobbies. A range of ME/CFS symptoms and comorbid health conditions further complicated management of ME/CFS. While some individuals learned to effectively manage their ME/CFS symptoms, most noted that available treatments provided limited relief.

**3.2.4. Functional impairments.** Several entries described progressive functional decline, culminating in becoming bedbound with severe symptoms:

*"For the first five years, he had a slow onset of CFIDS/ME until he became totally bedridden and got a diagnosis. When he had to be in a hospital when he was unable to eat or drink, the doctors and nurses had no training in this illness. When he returned home after more than one period of death that he was resuscitated from, his main carer was his mother who also has CFIDS/ME. He remained bedbound until his death"* [25].

*"He had four vaccines that were the trigger to allow CFIDS/ME to take over his life for the next three years. (He) used to play the saxophone and many sports as well as earning a brown belt in karate. That was replaced by flu-like symptoms, nausea, pain and fevers… His symptoms became so intense that just resting his head on a pillow was not possible"* [25].

The suffering and severe limitations that some with ME/CFS experienced sometimes led to viewing death as a form of relief:

*""When you read this I am at rest, free from suffering at last." She was "homebound and mostly bedbound" as her symptoms worsened dramatically to the point where she could no longer allow her friends and relatives to visit"* [25].

*"(She) died by her own hand by ingesting an overdose... Her partner… took his own life a year later. (She) had been diagnosed with ME in (country) and had been bed bound and used a wheelchair. She had been severely ill for a long time. Her inquest revealed she had left a suicide note and had persuaded her partner to respect her wishes"* [25].

Severe symptoms of ME/CFS sometimes limited the ability to live independently, leading individuals to move in with caregivers, often their family members:

*"After several doctors misdiagnosed her, she was finally diagnosed correctly but her ME worsened until she had to move back home with her parents"* [25].

*"A clinic in (city) diagnosed her… with a diagnosis of "chronic fatigue syndrome", had to move in with her father. A year later, (she) found she was in paralysis from her neck down. Her father asked for a nursing help while (she) remained mostly bedbound and unable to sleep"* [25].

Others who maintained employment still experienced significant limitations, with some noting job loss:

*"He lost all but two of his jobs due to poor health until he finally received disability benefits yet all who knew him could recognize his high intelligence"* [25].

*"(She) had to give up her job as a music teacher when her ME became worse"* [25].

*"He was in his early 40s and had lost his job, wife, and family due to his illness"* [25].

*"She was hired and was working but her illness made her miss many days of work until she was fired"* [25].

The visible physical toll of ME/CFS was sometimes notable. One entry quoted a journalist describing witnessing the condition:

*"I have seen African children suffering from starvation, met people dying of AIDS, patients paralysed from the neck down, others in the last stages of terminal cancer, but I had never seen a living person as desperately ill as (her)"* [25].

**3.2.5. Illness experiences.** Entries described a range of symptoms and severity levels among individuals with ME/CFS.
Some individuals engaged in extensive efforts to manage ME/CFS symptoms, often with little success:

*"She tried many different methods to see if they would help, including nutritional protocols"* [25].

*"He was on an experimental trial of Ampligen but found it was not a help and had adverse effects"* [25].

 

Symptoms noted included profound exhaustion, sensory sensitivities, gastrointestinal disturbances, and neurological problems. Pain was also particularly notable among entries:

*"(He) suffered from extreme pain in addition to numerous other symptoms and social isolation"* [25].

*"A stay at a hospital only worsened him and even morphine could not alleviate his pain"* [25].

Comorbid conditions such as fibromyalgia and cancer further complicated diagnosis, treatment, and quality of life:

*"Wrote a friend, "Sometimes her chemo and cancer pain screamed louder than her CFIDS/ME but it was always there. On the days when most cancer patients would have been able to get out or feel better, (she) had the double duty of then dealing with her ME""* [25].

*"She had a severe and long battle with CFIDS/ME and fibromyalgia when she sought help to end her life in June of 1997"* [25].

Some individuals described learning to manage their symptoms:

*"During that decade, he learned how to manage his life through nutrition and rest"* [25].

However, many others tried a host of treatments, with little success:

*"After careful researching, she tried various methods of treatment but found none helpful"* [25].

*"(He) was a professional photographer who traveled to many countries to try treatments and said he "suffered horrendous neglect since I got sick""* [25].

*"For the last several years of her life, (she) explored every possibility to find an effective treatment for CFIDS, but always met with a stone wall"* [25].

**3.2.6. Life satisfaction.** Entries reported the substantial impact of ME/CFS on quality of life, with many individuals reporting significant emotional distress. Financial and legal burdens such as medical expenses, limited occupational abilities, health insurance issues, and barriers to disability benefits were noted as furthering hardship:

*"(He), a long-term patient, took his own life… to escape the poverty, pain, and legal hassles brought on by this illness"* [25].

The stigma faced by many individuals with ME/CFS was another substantial burden:

*"The guilt, shame, and suggestion of others that there was nothing wrong were destroying me more than the illness"* [25].

Grief was common, with entries describing the loss of a planned future and personal autonomy:

*"She said, "For five years, I have been living in hell." She missed so much of what she dreamed of doing in the future and hung on until she could no longer endure her tortuous life of ME and subsequently asked for assisted suicide"* [25].

Others described the pain of being left behind as life moved on for others:

*"Her own words were spoken at her funeral: "It is not much wonder sufferers become depressed when they see the whole world passing them by. Life is going on without them. it is like trying to catch as escalator that is just beyond reach""* [25].

Suicide was frequently noted as a cause of death. In some instances, it was described as a response to an anticipated loss of independence, including the possibility of being placed in a nursing home with staff who may be uneducated regarding working with people with ME/CFS:

*"Living alone,* (she) *worsened and realized she could no longer care for herself. A nursing home seemed inevitable. She decided to take the only other option and end her life"* [25].

Other individuals expressed the feeling that ME/CFS had already taken their lives:

*"She said, "My life has become an inhumane existence." She left a husband and two teenaged children whom she referred to when she said, "I believe they have lost their mother to ME""* [25].

Several entries noted a sense of finality, with individuals feeling they had nothing left to give:

*"Her last message to all of us was, "Miss me but let me go""* [25].

*"She once told a fellow patient that she had "cried all the tears out of her... she didn't know if she had any more left""* [25].

One entry included a line from a woman's suicide note, directed towards her physician:

*"The documentary has her mother reading aloud a portion of her suicide note addressed to her physician where she said, "Maybe today will be tolerable""* [25].

A visual representation of the emerging themes and their relationship is shown in Fig 1.

This figure is a visual representation of the interconnected layers of the ME/CFS experience. Each layer represents a main theme within our findings, with the individual as well as their personal burdens and quality of life, at the core. Expanding outward is the social disconnection and advocacy theme, which is largely representative of the micro- and meso- systems of Bronfenbrenner's' Ecological Model. Next, is the clinical neglect and failures theme which aligns with both micro- and exo- systems of the ecological model. Lastly, the outermost layer is representative of systemic neglect and institutional failures which most closely mirrors the macrosystem in the ecological model. Identified subthemes are shown in bolded text.

## Discussion

The memorial entries of individuals with ME/CFS reported substantial institutional obstacles to acknowledgement and appropriate financial investment. The entries described a struggle with adequate payment from insurance companies, which aligns with other work in ME/CFS [30,31]. Similarly, entries described struggles with obtaining legal disability status, fitting with other work detailing obstacles in obtaining disability status for people with ME/CFS [30]. ME/CFS has been substantially underfunded relative to the impact of the condition, and National Institutes of Health (NIH) funding for ME/CFS would need to increase by 40-fold to meet the burden of the condition [6]. The entries reported feelings of hopelessness associated with the lack of resources dedicated to ME/CFS, aligning with past work noting high levels of hopelessness among individuals with ME/CFS [14]. The entries described a sense of frustration emerging from interactions

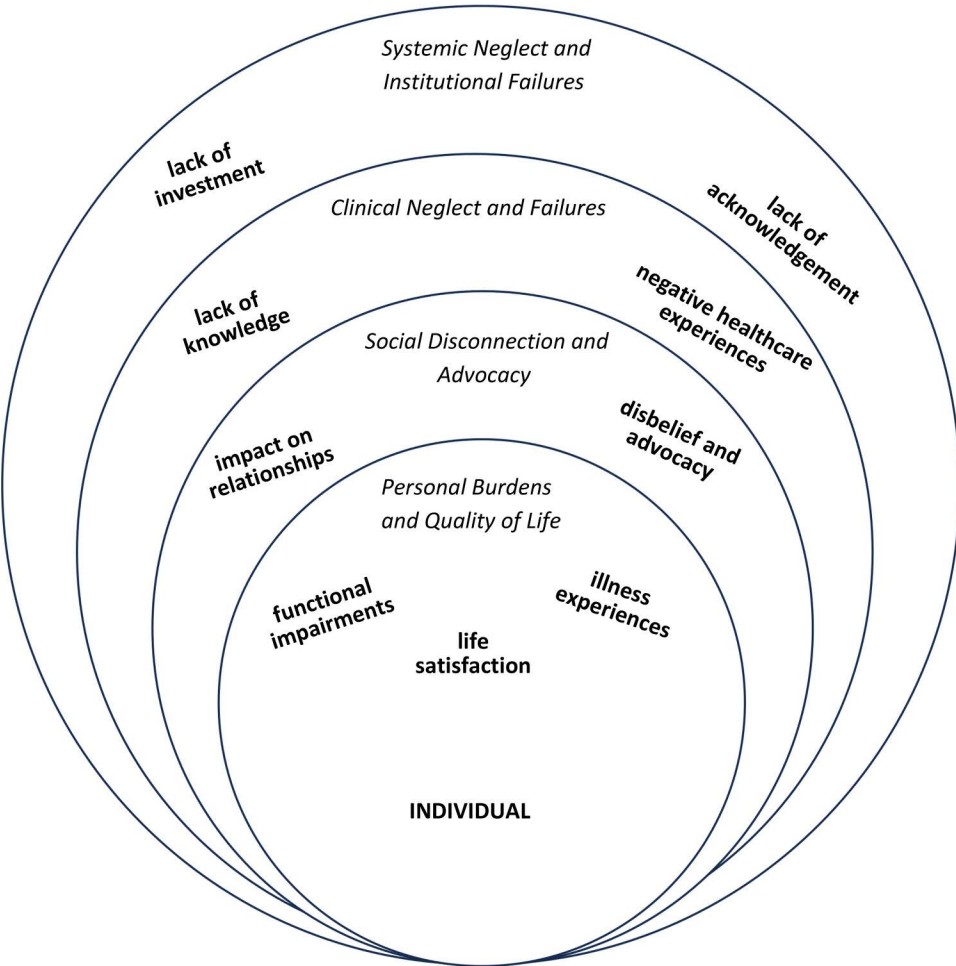

**Fig 1. A structural representation of emerging themes and their interrelationships.**

with larger systems, with effects that were perceived to trickle down to impact other environmental, social and personal experiences. For example, lack of research funding for ME/CFS from governmental agencies was perceived to impact the education of healthcare providers, subsequentially limiting effective care. The lack of acknowledgement and support from governmental bodies, insurance companies, and during the legal disability process affected the ability of individuals to maintain adequate financial status and employment, as well as contributed to seeking routes for advocacy to improve conditions. These findings highlight the significant impact of the macrosystem or larger societal structure and culture on an individuals' illness experience.

The systems level limitations contributed to issues noted at the clinical level. Entries frequently described poor healthcare experiences, including a history of prolonged misdiagnosis, misattribution of symptoms, dismissal of ME/CFS and more generally of individuals with ME/CFS themselves, and poor experiences with treatment. Each of these experiences has been supported in the larger ME/CFS literature [32–34]. Individuals with ME/CFS often experience years of misdiagnosis or underdiagnosis, with some estimates reporting that up to 90% of individuals with ME/CFS may be undiagnosed [32]. Dismissal of patients by healthcare providers has also been repeatedly reported, with patient reports often viewed as irrelevant, too emotional, or too time-consuming, perhaps exacerbated by a lack of resources available in understanding

and treating ME/CFS [33]. Unhelpful or harmful treatments being recommended for individuals with ME/CFS have also been previously reported [34]. Several treatments that have been historically recommended for ME/CFS, including graded exercise therapy and cognitive behavioral therapy, have since been suggested to be potentially harmful to individuals with the condition [34]. This emphasizes the potential for clinicians' lack of awareness of the latest ME/CFS research and recommendations to suggest harmful treatments to individuals with ME/CFS, with negative impacts. Information regarding ME/CFS such as diagnostic criteria and effective treatments are often not systematically taught at medical training institutions [35], increasing the likelihood that clinicians may not only have limited prior experience but also no training to care for individuals with ME/CFS. The prevalence and lasting impact of these negative healthcare experiences, characterizes the significant impact of the ecological model's microsystem and exosystem, or direct and indirect societal systems on the illness experiences of individuals managing ME/CFS. This dissatisfaction with healthcare experience may further be explained by prior research on explanatory models, or the patients' individual understanding of their illness [36]. This research emphasizes the importance of healthcare professionals seeking an understanding of the patient's individual conceptualization, beliefs, and cultural understanding of their illness, to improve treatment plans, patient-provider communication, and other outcomes [36–38]. Participant report in this study highlights a lack of explanatory model usage by clinicians, in which patient input is dismissed or met with disbelief, which may contribute to the poor outcomes and patient-provider relations described above.

Individuals with ME/CFS also reported issues with social relationships due to their condition, resulting in feelings of isolation and loss of relationships. Social isolation has been well documented in the context of ME/CFS, with disbelief of significant others associated with feelings of lack of support, frustration, and fear [14,39]. Loneliness has been noted among individuals with ME/CFS, with loss of independence, loss of function, and feelings of being left behind contributing to negative mental health outcomes [40]. Several entries in the memorial list described divorce and the ending of long-term relationships after the onset of ME/CFS, aligning with other work noting the breakdown of family relationships among individuals with chronic illness [41]. Entries frequently described not only a lack of competent care from healthcare providers, but also being dismissed by family members and close acquaintances. Disbelief from others can influence the mental health of individuals with ME/CFS, with not feeling understood by others suggested as a contributing factor to suicidal thoughts [14]. These challenges with dismissal and disbelief from others point to a common experience of those managing chronic illness: stigmatization. The concept of stigma has been broadly researched and can contribute to an altered sense of identity and social ostracism due to negative stereotypes for conditions such as chronic illness, as explored by stigma theory [42]. Limited research has been conducted to explore the impact of stigma that lacks visibility, including chronic illness, however, one study proposes the importance of understanding this stigma through a broader socio-cultural framework to highlight the health disparities of those with concealable stigma [43]. Further research on stigma within concealable chronic illness found internalized and anticipated stigma from others to significantly and negatively impact an individual's self-concept, or how they view themselves as someone with chronic illness [44]. This emphasizes the multilevel, internal and external, impact of stigma on physical and psychological health outcomes [45]. Our findings may further accentuate the need for a broader systems level approach in understanding the effects of social stigmatization in those with ME/CFS.

Given the experience of lack of understanding and support from those close to them, some with ME/CFS reported turning to advocacy opportunities or support groups. Online support groups have been perceived to reduce perceived feelings of depression and increase quality of life among individuals with ME/CFS [46]. Several benefits of support group membership have been identified among individuals with ME/CFS and fibromyalgia, a closely related chronic pain condition, including legitimization of the condition, providing new helpful information, and feeling understood by others [47]. Approximately 80% of individuals associated with a support group and 97% of active members of a support group reported support group membership to be generally helpful [47]. Given the perceived stigma and disbelief among personal connections and healthcare providers, individuals with ME/CFS reported turning to advocacy to improve perceptions of their condition

and campaign for more research funding and improved treatments. This presence and/or lack of social relationships within the narratives provides a representation of the ecological model's microsystem, in which an individual is directly affected by their direct social connections in both positive and negative ways. Additionally, we also observed the positive impact of broad social support, particularly through interactions between microsystems, which represents the mesosystem of the ecological model (i.e., multiple family members engaging in advocacy efforts, or family engagement with healthcare professionals).

Individuals with ME/CFS also reported significant impacts of ME/CFS on functional abilities and quality of life, with some entries noting direct associations between quality of life and death, particularly in deaths by suicide. Entries noted loss of independence and needing to move in with others, often family members. Others noted substantial impairments related to activities of daily living or employment activities. Each of these limitations has been previously reported in the literature, with individuals with ME/CFS experiencing a range of functional limitations [48]. The dependency noted by individuals with ME/CFS can also impact those who care for them, with over 70% of carers for individuals with severe ME/CFS endorsing providing more than 40 hours per week of care, altering the carer's work, social, and financial situations [49]. Up to 75% of individuals affected by ME/CFS are unable to work, with approximately 25% consistently housebound or bedbound, and the financial impacts of the condition both at the societal and individual level can be significant [13,50]. The entries from the memorial list contextualized these findings, noting financial stress and dependency on others, with some noting moving into another's home as independence declined. Substantial proportions of individuals with ME/CFS report feeling hopeless for future relief, and it has been estimated that only approximately 5% of those with ME/CFS fully recover [51]. Perhaps due to these barriers, individuals described in the memorial list sometimes turned to advocacy, such as engagement with social media or writing blogs about ME/CFS, participating in healthcare and governmental education regarding ME/CFS, and donating financial resources and even their bodies for medical research following death. These individual impacts and experiences highlight the individual at the core of their own experience, as referenced by the ecological model with the individual as the foundation with surrounding external influences.

Memorial entries also described the impact of comorbid conditions and particularly limiting symptoms, with many mentioning chronic pain as a highly prevalent and impairing symptom. Chronic pain has been noted to be significant among other samples with ME/CFS [52,53], and the memorial list narratives suggest that pain may be one of the most burdensome symptoms among affected individuals. Several entries described the wearing nature of chronic symptoms, with several individuals enduring repeated misdiagnoses. Delay in the diagnosis of ME/CFS has been associated with lower odds of recovery or improvement [54], indicating that even individuals reporting chronic ME/CFS may have experienced symptoms for longer than their formal diagnosis. This delay in diagnosis may complicate an individual's ability to accept and take ownership of a chronic illness identity, which some have suggested as an important step in managing and coping with chronic illness [55]. While some entries in the memorial list noted successful management of symptoms, others reported little to no improvement or even worsening despite repeated trials of treatments and a long list of treating healthcare providers.

Entries frequently described mental health concerns and poor life satisfaction due to their symptoms and functional limitations. Entries reported financial stressors, dependence, and social isolation as contributors to mental health symptoms. Depressive and anxiety symptoms have been frequently noted in the context of ME/CFS with notable proportions of individuals experiencing clinically significant symptoms [56,57]. Health-related quality of life among individuals with ME/CFS is significantly lower than the population mean and the lowest reported health-related quality of life score compared to individuals with several other health conditions, including stroke, cancer, diabetes, and heart attacks [16,18]. The substantial challenges faced by individuals with ME/CFS have been suggested to influence suicide risk [58], and our analysis supports this conclusion. Suicide has been suggested to be among the most frequently reported causes of death among individuals with ME/CFS [21–23]. Individuals with ME/CFS have been suggested to be over six times more likely to die by suicide compared to the general population [59]. The entries on the memorial list illustrated the contexts in which

individuals died by suicide, often noting contributing factors such as hopelessness, pain, social isolation, loss of independence, and dismissal or poor treatment by healthcare providers. This emphasizes the immense impact of ME/CFS as an illness, and the layers of social and societal impact that influence the lived experience of each individual.

The frequent mention of suicide in the narratives is also notable, particularly when considered in the context of similar poorly understood and stigmatized conditions such as chronic pain. Several factors are suggested to contribute to suicidality in the context of chronic pain, including impaired feelings of belonging and feeling like a burden as suggested in the interpersonal theory of suicide in chronic pain [60]. Mental defeat has also been suggested as a contributor to suicide in the context of chronic pain [61], which may parallel feelings of hopelessness noted in ME/CFS [40]. Indeed, both hopelessness and chronic pain have been independently associated with higher risk of screening positive for suicide risk [62]. The "ideation-to-action" framework hypothesizes that pain and hopelessness combine to contribute to suicidal ideation [63]. While pain in this model is traditionally understood as psychological in nature, physical and psychological pain are tightly connected [64], suggesting that hopelessness and chronic pain may additively contribute to suicide risk. Particularly considering the frequent mentions of chronic pain in the analyzed memorial narratives, frameworks developed to understand suicidality in the context of chronic pain may also provide useful insight for better understanding suicidality in ME/CFS.

The memorial nature of the narratives analyzed is also notable. Memorials have been suggested to provide meaning to acquaintances of the deceased, while also suffering from a lack of control over how emotions and memories that are shared in memorial narratives are framed [65]. The potential power of memorial narratives in the context of ME/CFS must be considered as well. Associated with significant stigma and dismissal [66,67], individuals in the ME/CFS community have turned to advocacy, as noted in this qualitative analysis. Others have suggested the role that shared grief and loss can have in shaping movement solidarity and calls for justice [68]. Similarly, activism has been suggested to offer a meaning making function for individuals experiencing ambiguous losses [68], which are common in the lived experience of ME/CFS [8,40], and may continue to affect acquaintances even after the individual's death. Considering these factors, the memorial records analyzed may provide a unique combination of both grief and activism in the context of ME/CFS.

There were several limitations to the present study. All data were self-reported by acquaintances of the deceased, and the accuracy of the entries depends on the interpretation and perceptions of the acquaintance reporting the memorial information to the National CFIDS Foundation. The analyses relied on existing publicly available data, and secondary data analyses can have several limitations including the original data not being collected for the secondary study and reliance on the data quality of the original source, which may introduce bias or misinterpretation [69]. Another important consideration is selection bias, as those placed on the memorial list may have had increased awareness or involvement with the National CFIDS Foundation, or other advocacy or support groups. Similarly, individuals involved in advocacy efforts may have been more likely to be listed on the memorial list, potentially limiting the generalizability of the findings. Individuals with more severe ME/CFS may be more likely to be listed on the memorial list, as some findings have suggested that individuals with more severe symptoms have been suggested to be more involved in self-advocacy [70].

Most participants were associated with North America [23], thus limiting our understanding of differences in the experience of those with ME/CFS in other areas of the world. Additionally, most deaths reported on the memorial list occurred between 2000s and 2019 [23], and it is unclear how the experience of people with ME/CFS may differ during other periods of time. It is unclear what diagnostic criteria, if any, the individuals listed on the memorial list may meet and medical and death records were not available for confirmation of cause of death or personal characteristics of the individuals listed. Despite these limitations, this thematic analysis offers insight into the nuances of those experiencing ME/CFS, providing unique insight into acquaintances' perceptions of the deceased individual's life and death in the context of ME/CFS. This study fills a gap in the literature by offering a perspective of the lived experience and death of individuals with ME/CFS from the view of those close to them. This work also provides qualitative insight and nuance to prior quantitative research exploring mortality in the context of ME/CFS [22,23].

Individuals with ME/CFS experience significant challenges at several environmental levels. From a systemic and institutional perspective, individuals with ME/CFS described on the memorial list face challenges in lack of acknowledgement and funding from insurance and disability authorities, as well as a lack of research funding from federal health agencies. This finding aligns with findings in the broader ME/CFS literature, finding lack of research funding and struggles with disability and insurance providers prominent frustrations among individuals living with ME/CFS [30,31]. Barriers are also experienced within the medical system and clinical interactions, with misdiagnosis and misattribution of symptoms common, as well as dismissal and poor treatment recommendations. Medical providers often lack education and experience related to ME/CFS, possibly contributing to poor management of the illness [35]. The memorial entries also emphasized misattribution and disbelief within personal relationships, furthering feelings of social isolation. Each of these levels of challenges faced by individuals with ME/CFS can contribute to a complicated illness experience characterized by impairing symptoms, chronicity, financial stress, poor social support, poor quality of life and physical and mental health impairment. Some entries described that the experience of challenges at these various environmental and larger systems levels compound, highlighting the complex nature of living with ME/CFS. Among some individuals, the barriers faced contributed to hopelessness and suicidality, variables also noted in other ME/CFS literature [71].

Given the complex nature of ME/CFS and the mortality risk of the illness highlighted in this study, it will be important for future research to further explore the experience of this illness through the lens of the ecological model or systems theory more broadly. Further research may expand on specific experiences and systems that contribute to mortality risk, specifically mortality by suicide. This conceptualization may also have clinical implications for those treating patients with ME/CFS, in understanding the interconnected and confounding impact of various systems on the experience, treatment, and outcomes of those with ME/CFS. Overall, our findings suggest that a more nuanced and comprehensive framework, such as an ecological model, is needed to thoroughly and more completely understand the lived experiences as well as address the health disparities and barriers faced by individuals with ME/CFS.

The acquaintances of deceased individuals with ME/CFS shared a complex and often vivid display of the experience of individuals with ME/CFS, focusing on death and perceived contributing factors. Some shared hope and remembrance of the progress that the deceased individuals sought, while others shared anguish, grief, and anger at the systems that they perceived as contributing factors to the lived experience and often death of individuals with ME/CFS. To illustrate that grief and loss in the context of ME/CFS can be complex and conflicted, we draw on a father's reflection on his daughter's death:

*"Her father … was quoted saying his daughter lived with "intractable and unrelenting pain" and, though he certainly was not happy to see his only child die, "There are things in this world worse than death""* [25].

## Supporting information

**S1 Table. Coded memorial entries.**
(XLSX)

**S2 Table. Codes and subcodes.**
(XLSX)

## Author contributions

**Conceptualization:** Zoe Sirotiak.

**Formal analysis:** Zoe Sirotiak, Hailey J Amro.

**Methodology:** Zoe Sirotiak.

**Writing – original draft:** Zoe Sirotiak, Hailey J Amro.

**Writing – review & editing:** Zoe Sirotiak, Hailey J Amro.

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
