## [Decision Letter · Decision Letter 0]

30 Dec 2025

PONE-D-25-44193Investigating the ME/CFS experience through qualitative analysis of memorial entriesPLOS One

Dear Dr. Sirotiak,

Thank you for submitting your manuscript to PLOS ONE. After careful consideration, we feel that it has merit but does not fully meet PLOS ONE’s publication criteria as it currently stands. Therefore, we invite you to submit a revised version of the manuscript that addresses the points raised during the review process.

We look forward to receiving your revised manuscript.

Kind regards,

Tanja Grubić Kezele, Ph.D., M.D.

Academic Editor

PLOS One

Journal Requirements:

**Additional Editor Comments:**

Based on the reviewers' suggestions, the paper needs a major revision. The reviewers' comments can be found below.

Reviewers' comments:

Reviewer's Responses to Questions

**Comments to the Author**

1. Is the manuscript technically sound, and do the data support the conclusions?

Reviewer #1: Partly

Reviewer #2: Yes

Reviewer #3: Yes

2. Has the statistical analysis been performed appropriately and rigorously? 

Reviewer #1: N/A

Reviewer #2: Yes

Reviewer #3: Yes

3. Have the authors made all data underlying the findings in their manuscript fully available?

Reviewer #1: Yes

Reviewer #2: Yes

Reviewer #3: Yes

4. Is the manuscript presented in an intelligible fashion and written in standard English?

Reviewer #1: Yes

Reviewer #2: Yes

Reviewer #3: Yes

5. Review Comments to the Author

Reviewer #1: Overall Assessment

This manuscript presents a thoughtful and emotionally resonant qualitative study exploring the lived experiences of individuals with myalgic encephalomyelitis/chronic fatigue syndrome (ME/CFS) as represented through 505 memorial entries on the National CFIDS Foundation website. Using inductive thematic analysis, the authors identify four interconnected domains—systemic neglect and institutional failure, clinical neglect and failures, social disconnection and advocacy, and personal burden and quality of life. The manuscript contributes novel qualitative insights into the psychosocial and systemic dimensions of ME/CFS-related mortality.

The paper is generally well organized, conceptually grounded, and written in clear academic prose. The authors appropriately situate their work within prior ME/CFS research and employ a theoretically sound analytic framework (reflexive thematic analysis within an ecological systems perspective). However, the manuscript would benefit from greater methodological transparency, theoretical integration, and refinement of the discussion to enhance its scientific rigor and interpretive depth.

Major Comments

1. Methodological Rigor and Transparency

Sampling and Data Source:

The use of publicly available memorial entries is innovative but introduces potential selection and interpretive bias. The authors acknowledge this briefly but should elaborate on how such memorials might disproportionately represent more severe cases or those with greater advocacy engagement. A short section on data provenance and limitations of second-hand accounts would strengthen credibility.

Analytic Process:

The manuscript describes inductive coding and theme generation but omits sufficient detail on the codebook development, reflexivity, and researcher positionality. Reflexive thematic analysis (per Braun & Clarke) emphasizes these aspects. The authors should specify:

How disagreements between coders were resolved beyond “discussion.”

Whether NVivo or another tool was used for coding.

How saturation was assessed.

Any steps taken to enhance analytic trustworthiness (e.g., audit trail, peer debriefing).

Ethical Considerations:

While the dataset is public, ethical discussion could be expanded. The authors might comment on the potential sensitivity of analyzing memorial narratives and their decision to retain pronouns but omit names.

2. Theoretical Framing

The manuscript references an ecological systems approach (Bronfenbrenner, 1977) but does not operationalize it clearly. Consider explicitly mapping findings to the microsystem, mesosystem, exosystem, and macrosystem levels rather than using the four-level structure as a loose analogy. The analysis could benefit from integrating sociological theories of stigma, chronic illness identity, and medical mistrust to contextualize systemic and clinical neglect (e.g., Goffman’s stigma framework, Kleinman’s explanatory models).

3. Interpretation and Discussion

Novelty and Contribution:

The study expands upon prior quantitative mortality work (e.g., Jason et al., 2006; Sirotiak & Amro, 2025). However, the discussion often repeats descriptive findings rather than advancing interpretive insights. The authors could discuss how memorial discourse itself functions—both as a site of grief expression and as activism within patient communities.

Causality and Attribution:

Some memorial narratives attribute deaths to ME/CFS or associated conditions. While these perceptions are meaningful, the paper should more clearly distinguish perceived from documented cause of death to avoid overstating medical conclusions.

Suicide Contextualization:

The rich qualitative material on suicidality could be expanded with references to chronic pain–related suicide frameworks and hopelessness models to connect individual narratives to broader psychosocial theories.

4. Presentation and Structure

The Results section is comprehensive but overly lengthy. Condensing exemplar quotations and focusing on the interpretive synthesis rather than repetition would improve readability.

The Figure 1 conceptual model is valuable; however, it should be simplified visually and accompanied by a concise explanation linking each thematic layer to Bronfenbrenner’s systems.

The Abstract slightly overstates the analytical rigor (“inductively developing a codebook” and “themes emerged within four societal levels”). Consider clarifying that this is a reflexive thematic analysis of public memorial narratives rather than primary participant data.

Minor Comments

Line 79–82: “underfunded and under-researched” — include citation to NIH funding statistics (e.g., Dimmock et al., 2020) for context.

Line 138: “Interrater reliability was not assessed” — briefly justify this decision per Braun & Clarke (2021) guidance.

Ensure consistent formatting of “CFIDS/ME” versus “ME/CFS.”

Typographical and stylistic issues: remove redundant spaces, align reference formatting with PLOS ONE style.

Consider shortening the Acknowledgements section—“The authors have no acknowledgements” is unnecessary; simply omit.

Recommendation

Recommendation: Major Revision

This manuscript offers a valuable qualitative perspective on mortality and lived experience in ME/CFS, addressing an underexplored area of research. However, to meet PLOS ONE’s standards for qualitative rigor and interpretive depth, the authors should strengthen methodological transparency, theoretical grounding, and critical interpretation. With revision, this paper could make a meaningful contribution to the literature on ME/CFS, health inequities, and chronic illness experiences.

Reviewer #2: Abstract

Additional symptoms could be included to better capture the breadth of experiences described.

Introduction

This is a well-written section. The authors may wish to consider integrating the following reference to further strengthen the context:

Thornton EJ, Hayes LD, Goodwin DS, Sculthorpe N, Prior Y, Sanal-Hayes NEM. Managing Energy, and Shaping Care: Insights from Adults with Myalgic Encephalomyelitis/Chronic Fatigue Syndrome Through Co-Production Workshops. Am J Med. 2025 Jun;138(6):1001–1009. doi:10.1016/j.amjmed.2025.02.008.

Materials and Methods

This section is well written and requires no revisions.

Results

In the opening paragraph, clearly outlining the main themes and subthemes would help guide the reader. The theme titled “Systemic neglect and institutional failure” appears to lack clarity regarding its structure. It is unclear whether this is intended as a main theme, with elements such as “lack of acknowledgement” presented as subthemes. Clarifying this hierarchy would improve readability for all sections.

Discussion

The first paragraph should briefly outline the key themes and discuss the main findings, rather than reiterating the introduction or the study’s significance, which should already be clear.

Line 538 requires additional references.

Lines 638–659 also require supporting references. Additionally, including participant excerpts in the discussion section is not appropriate, instead, these excerpts should be interpreted and discussed.

Reviewer #3: This study conducted a thematic analysis of the memorial entries from the National CFIDS Foundation memorial list submitted by family, friends, or acquaintances following the deaths of individuals living with CFS/ME. The data for the study are publicly available National CFIDS Foundation memorial list.

The manuscript is well-written, well structured, clear in reasoning, premises and conclusions. The writing clearly lays out and justifies the aims of research, the justification, the nature of the methodology applied, and the limitations. The authors have provided detailed descriptions of the themes identified from the memorial entries, and explained their reasoning process, and provided data to back up their conclusions.

It might be informative for the readers if the authors included a coding manual, similar to previous thematic analysis studies, (e.g. Picariello, F., Ali, S., Moss-Morris, R., & Chalder, T. (2015). The most popular terms for medically unexplained symptoms: the views of CFS patients. Journal of psychosomatic research, 78(5), 420-426.) to help present methodological steps in a concise and clear way.

6. PLOS authors have the option to publish the peer review history of their article (what does this mean? ). If published, this will include your full peer review and any attached files.). If published, this will include your full peer review and any attached files.

**Do you want your identity to be public for this peer review?** For information about this choice, including consent withdrawal, please see our For information about this choice, including consent withdrawal, please see our Privacy Policy ..

Reviewer #1: No

Reviewer #2: No

Reviewer #3: No

---

## [Author Response · Author response to Decision Letter 1]

18 Jan 2026

Dear Dr. Grubić Kezele,

We are submitting our revised manuscript “Investigating the ME/CFS experience through qualitative analysis of memorial entries” for reconsideration for publication in PLOS One. We have responded to each reviewer comment below. The authors thank the editor and reviewers for the careful review and thoughtful comments.

Reviewer #1: Overall Assessment

This manuscript presents a thoughtful and emotionally resonant qualitative study exploring the lived experiences of individuals with myalgic encephalomyelitis/chronic fatigue syndrome (ME/CFS) as represented through 505 memorial entries on the National CFIDS Foundation website. Using inductive thematic analysis, the authors identify four interconnected domains—systemic neglect and institutional failure, clinical neglect and failures, social disconnection and advocacy, and personal burden and quality of life. The manuscript contributes novel qualitative insights into the psychosocial and systemic dimensions of ME/CFS-related mortality. The paper is generally well organized, conceptually grounded, and written in clear academic prose. The authors appropriately situate their work within prior ME/CFS research and employ a theoretically sound analytic framework (reflexive thematic analysis within an ecological systems perspective). However, the manuscript would benefit from greater methodological transparency, theoretical integration, and refinement of the discussion to enhance its scientific rigor and interpretive depth.

Thank you for your thoughtful review.

Major Comments

1. Methodological Rigor and Transparency

Sampling and Data Source:

The use of publicly available memorial entries is innovative but introduces potential selection and interpretive bias. The authors acknowledge this briefly but should elaborate on how such memorials might disproportionately represent more severe cases or those with greater advocacy engagement. A short section on data provenance and limitations of second-hand accounts would strengthen credibility.

Thank you for this note. We agree that selection and interpretive bias is important in this study and we have expanded on this possibility in our limitations. We have acknowledged that individuals with greater levels of advocacy as well as more severe symptoms may be particularly likely to be listed on the memorial list. In addition, individuals involved with the National CFIDS Foundation, the foundation that manages the memorial list, or other support groups or advocacy organizations may be more likely to be listed as well. We have expanded on these possibilities in our Discussion section:

“The analyses relied on existing publicly available data, and secondary data analyses can have several limitations including the original data not being collected for the secondary study and reliance on the data quality of the original source, which may introduce bias or misinterpretation (69). Another important consideration is selection bias, as those placed on the memorial list may have had increased awareness or involvement with the National CFIDS Foundation, or other advocacy or support groups. Similarly, individuals involved in advocacy efforts may have been more likely to be listed on the memorial list, potentially limiting the generalizability of the findings. Individuals with more severe ME/CFS may be more likely to be listed on the memorial list, as some findings have suggested that individuals with more severe symptoms have been suggested to be more involved in self-advocacy (70).” (pages 34, lines 702-711)

Analytic Process: The manuscript describes inductive coding and theme generation but omits sufficient detail on the codebook development, reflexivity, and researcher positionality. Reflexive thematic analysis (per Braun & Clarke) emphasizes these aspects.

Thank you for this comment. We have expanded our Methods section to address codebook development and reflexivity. We have also added a research positionality section that describes the background and perspective of each researcher. These sections of the Methods now read:

“Coders also identified areas of confusion or lack of clarity in their own coding framework to prepare for codebook finalization in discussion. The coders then met to establish a codebook, discussing the meanings and distinctions of the codes developed. Coders shared their code list and framework, exploring similarities and differences between their coding frameworks. Discussion between coders included identifying foundational codes based on concepts that were expressed repeatedly, as well as subcodes. The coders also explored how and why interpretations of some codes may differ between coders or be unclear, resulting in clarifying and defining these codes. The coders engaged in reflexivity throughout the coding and thematic analysis process, examining their assumptions and potential biases. The coders considered how their educational background, professional training, personal experiences, and underlying assumptions may have influenced their interpretation of the memorial narratives and resulting code development. Discussion of themes between the coders was designed to allow consideration of diverging perspectives, resulting in a more comprehensive understanding of the narratives. Saturation was not assessed as this was a secondary data analysis. Saturation has been criticized as a heterogenous and inconsistently applied concept (28), further supporting its omission from this analysis.

Each of the coders independently coded each of the entries before meeting again to discuss emerging initial themes. Each code was dichotomously coded as either “present” or “absent” within each memorial narrative utilizing an Excel document. Examples of these coded memorial entries are shown in Supplementary Table 2, with specific participant memorials removed for privacy purposes. The coders identified associations between codes and resulting themes, as well as associations between themes. Interrater reliability was not assessed, aligning with recommendations for thematic analyses (29). Instead, the coders considered individual interpretations and worked together to develop themes that considered the understanding of both coders. Analytic trustworthiness was promoted through coder discussions in which coders compared their interpretations and reflected on how their background and perspective may influence coding decisions and conclusions. After these themes were identified and named, the researchers read through selections of the entries to ensure that these themes accurately and adequately described the ideas and experiences communicated in the entries. The themes were then finalized with descriptive quotes identified to illustrate the emerging themes. Emerging themes and subthemes are displayed in Table 1, and codes and subcodes are shown in Supplementary Table 1.

While all entries analyzed are publicly available (25), full names have been omitted to protect privacy. All quotes are provided verbatim, including original spelling, punctuation, and grammatical errors, to allow for the voice of the individual or their acquaintance to be accurately shared. Although these data were publicly available, memorial narratives are inherently sensitive and personal, and we approached our analyses with care. We chose to remove names and other specific information (i.e., city, insurance company name) to reduce direct identifiability while retaining gendered pronouns (he/she) to allow narrative coherence and respecting the expressive purpose of the original text. (pages 7-10, lines 148-189)

“Positionality

The authors acknowledge the impact of their identities and experiences in shaping their interpretations and conclusions. The first author is a white, cisgender American woman. She holds a bachelor’s degree in kinesiology and psychology, a Doctor of Physical Therapy degree, a master’s degree in kinesiology, and is pursuing a PhD degree with majors in immunobiology, kinesiology, and psychology. Her academic background informs her focus on the interconnections between physical and mental health in the context of chronic health conditions, seeing all health conditions as not dichotomously physical or mental health issues. She acknowledges her social privilege due to her education and academic training, and she appreciates her lived experience limitations regarding personal experience of underprivileged social identities. She has lived with ME/CFS for approximately six years, contributing lived experience and a patient perspective that may influence how she interprets memorial narratives. The second author is a cisgender American woman with mixed Arab heritage. She holds a bachelor’s degree in psychology, a Master of Social Work degree, and is pursuing a PhD degree in Counseling Psychology. Her academic background and clinical social work experience inform her focus on an ecological systems framework, which emphasizes the impact of external factors on an individual's lived experiences. Her background also informs her focus on highlighting the lived experiences of and addressing the barriers faced by marginalized communities such as those with chronic illness. She acknowledges her social privilege stemming from her academic and social backgrounds, as well as her lack of personal experience in living with chronic illness. She made conscious efforts to check existing biases while interpreting the memorial narratives; however, her held identities may ultimately influence her interpretation of the data.” (pages 10-11, lines 191-211)

The authors should specify: How disagreements between coders were resolved beyond “discussion.”

Thank you for this comment. As suggested by Braun & Clarke, we did not assess interrater reliability. However, we engaged in discussion regarding our interpretations as well as codes that we were unsure about. We also reflected on how our experiences and academic background may have influenced our interpretations and conclusions. We have expanded on the development of the codebook and thematic analysis in the Methods section:

“A thematic analysis using inductive coding strategies (26,27) was performed to analyze entries on the memorial list. Two researchers, each with experience in coding and interpreting qualitative data, were involved in the coding process. First, the coders familiarized themselves with the data, reading through each of the entries and taking notes regarding potential codes as well as patterns demonstrated in the entries. Coders also identified areas of confusion or lack of clarity in their own coding framework to prepare for codebook finalization in discussion. The coders then met to establish a codebook, discussing the meanings and distinctions of the codes developed. Coders shared their code list and framework, exploring similarities and differences between their coding frameworks. Discussion between coders included identifying foundational codes based on concepts that were expressed repeatedly, as well as subcodes. The coders also explored how and why interpretations of some codes may differ between coders or be unclear, resulting in clarifying and defining these codes. The coders engaged in reflexivity throughout the coding and thematic analysis process, examining their assumptions and potential biases. The coders considered how their educational background, professional training, personal experiences, and underlying assumptions may have influenced their interpretation of the memorial narratives and resulting code development. Discussion of themes between the coders was designed to allow consideration of diverging perspectives, resulting in a more comprehensive understanding of the narratives.” (page 7, lines 144-161)

Whether NVivo or another tool was used for coding.

Thank you for this suggestion. We have clarified that each code was dichotomously coded as either “present” or “absent” for each individual memorial narrative within an Excel document. No qualitative methodological or analysis tool was utilized. We also added an additional table to our supplemental material to provide further clarification on our coding process. We have expanded on this approach in the Methods section:

“Each code was dichotomously coded as either “present” or “absent” within each memorial narrative utilizing an Excel document. Examples of these coded memorial entries are shown in Supplementary Table 2, with specific participant memorials removed for privacy purposes.” (page 8, lines 165-168)

How saturation was assessed.

Thank you for this comment. Saturation was not assessed because this study involved secondary data analysis, and thus we could not engage in primary sampling or data collection, which are central to how saturation has been traditionally evaluated. In addition, recent methodological work has criticized saturation as a heterogeneous and inconsistently applied concept across qualitative methodologies rather than a universal standard (Tight, 2023), indicating that its omission is a valid methodological decision. We have clarified this rationale in the Methods section:

“Saturation was not assessed as this was a secondary data analysis. Saturation has been criticized as a heterogenous and inconsistently applied concept (28), further supporting its omission from this analysis.” (page 7, lines 161-163)

Tight, M. (2023). Saturation: An overworked and misunderstood concept? Sage Journals, 30(7). https://doi.org/10.1177/10778004231183948

Any steps taken to enhance analytic trustworthiness (e.g., audit trail, peer debriefing).

While no formal procedures such as an audit trail were conducted, the research team enhanced analytic trustworthiness through collaborative discussions regarding their individual interpretations and how these interpretations may be biased or influenced by personal background and experiences. The coders compared and reflected on their separate interpretations of the data, considering how their individual perspectives and experiences might influence coding decisions and conclusions. This reflexive process helped to identify potential biases and promote more nuanced and credible analysis. We have now explained this process in our Methods section:

“Analytic trustworthiness was promoted through coder discussions in which coders compared their interpretations and reflected on how their background and perspective may influence coding decisions and conclusions.” (page 8, lines 172-174)

Ethical Considerations:

While the dataset is public, ethical discussion could be expanded. The authors might comment on the potential sensitivity of analyzing memorial narratives and their decision to retain pronouns but omit names.

Thank you for this important point. We have expanded on our decision to remove names but retain pronouns in our analysis, emphasizing our goal to balance privacy of individual experiences while also respecting narrative coherence and the expressive purpose of the original text. We have also now removed other specific information such as city and insurance company name to further reduce direct identifiability:

“While all entries analyzed are publicly available (25), full names have been omitted to protect privacy. All quotes are provided verbatim, including original spelling, punctuation, and grammatical errors, to allow for the voice of the individual or their acquaintance to be accurately shared. Although these data were publicly available, memorial narratives are inherently sensitive and personal, and we approached our analyses with care. We chose to remove names and other specific information (i.e., city, insurance company name) to reduce direct identifiability while retaining gendered pronouns (he/she) to allow narrative coherence and respecting the expressive purpose of the original text.” (pages 10, lines 182-189)

2. Theoretical Framing

The manuscript references an ecological systems approach (Bronfenbrenner, 1977) but does not operationalize it clearly. Consider explicitly mapping findings to the microsystem, mesosystem, exosystem, and macrosystem levels rather than using the four-level structure as a loose analogy. The analysis could benefit from integrating sociological theories of stigma, chronic illness identity, and medical mistrust to contextualize systemic a

---

## [Decision Letter · Decision Letter 1]

5 Feb 2026

Investigating the ME/CFS experience through qualitative analysis of memorial entries

PONE-D-25-44193R1

Dear Dr. Sirotiak,

We’re pleased to inform you that your manuscript has been judged scientifically suitable for publication and will be formally accepted for publication once it meets all outstanding technical requirements.

Kind regards,

Tanja Grubić Kezele, Ph.D., M.D.

Academic Editor

PLOS One

Additional Editor Comments (optional):

Reviewers' comments:

Reviewer's Responses to Questions

**Comments to the Author**

1. If the authors have adequately addressed your comments raised in a previous round of review and you feel that this manuscript is now acceptable for publication, you may indicate that here to bypass the “Comments to the Author” section, enter your conflict of interest statement in the “Confidential to Editor” section, and submit your "Accept" recommendation.

Reviewer #1: All comments have been addressed

Reviewer #2: (No Response)

2. Is the manuscript technically sound, and do the data support the conclusions?

Reviewer #1: Yes

Reviewer #2: Yes

3. Has the statistical analysis been performed appropriately and rigorously? 

Reviewer #1: Yes

Reviewer #2: Yes

4. Have the authors made all data underlying the findings in their manuscript fully available?

Reviewer #1: Yes

Reviewer #2: Yes

5. Is the manuscript presented in an intelligible fashion and written in standard English?

Reviewer #1: Yes

Reviewer #2: Yes

6. Review Comments to the Author

Reviewer #1: The authors have adequately and thoughtfully addressed all of my prior questions and comments. The revised manuscript is substantially strengthened, particularly with respect to methodological transparency, theoretical framing, and clarity of interpretation. The qualitative analytic approach is appropriate for the research aims, and the use of publicly available memorial narratives is handled with care, reflexivity, and appropriate ethical consideration.

This study makes a meaningful contribution to the ME/CFS literature by providing rich qualitative insight into lived experience, systemic and clinical barriers, and perceptions surrounding mortality—an area that remains underexplored despite growing quantitative evidence. The authors clearly articulate limitations inherent to secondary analysis of memorial data and avoid overstating causal claims, while still offering important interpretive insights with relevance for research, clinical practice, and health policy. The manuscript is well organized, clearly written, and theoretically grounded. Ethical considerations, including privacy protection and sensitivity to the nature of memorial narratives, are appropriately addressed. I have no concerns regarding dual publication, research ethics, or publication ethics. The work appears original, transparently reported, and compliant with journal standards. Based on the quality of the revisions and the overall rigor and contribution of the manuscript, I recommend this paper for publication.

Reviewer #2: Thank you for addressing all comments especially the integration of more literature and references.

7. PLOS authors have the option to publish the peer review history of their article (what does this mean? ). If published, this will include your full peer review and any attached files.). If published, this will include your full peer review and any attached files.

**Do you want your identity to be public for this peer review?** For information about this choice, including consent withdrawal, please see our For information about this choice, including consent withdrawal, please see our Privacy Policy ..

Reviewer #1: **Yes:** Patrick C. HardiganPatrick C. Hardigan

Reviewer #2: No

---

## [Editor Report · Acceptance letter]

PONE-D-25-44193R1

PLOS One

Dear Dr. Sirotiak,

I'm pleased to inform you that your manuscript has been deemed suitable for publication in PLOS One. Congratulations! Your manuscript is now being handed over to our production team.

Kind regards,

on behalf of

Prof. dr. Tanja Grubić Kezele

Academic Editor

PLOS One